# Anterior Mandibular Displacement in Growing Rats—A Systematic Review

**DOI:** 10.3390/ani12162059

**Published:** 2022-08-12

**Authors:** Efstratios Ferdianakis, Ioannis Lyros, Ioannis A. Tsolakis, Antigoni Alexiou, Konstantina Alexiou, Apostolos I. Tsolakis

**Affiliations:** 1Department of Orthodontics, School of Dentistry, National and Kapodistrian University of Athens, 11527 Athens, Greece; 2Department of Orthodontics, School of Dentistry, Aristotle University of Thessaloniki, 54623 Thessaloniki, Greece; 3Department of Oral and Maxillofacial Radiology, School of Dentistry, National and Kapodistrian University of Athens, 11527 Athens, Greece; 4Department of Orthodontics, Case Western Reserve University, Cleveland, OH 44106, USA

**Keywords:** mandibular protrusion, functional appliances, mandibular anterior displacement, rats

## Abstract

**Simple Summary:**

Mandibular deficiency is a very common orthodontic problem. Many different types of appliances have been utilized to correct such malocclusions. Most are appliances that alter the function of the mandible resulting in a more forward positioning of the lower jaw. Many researchers state that such an alteration results in a different rate of mandibular growth, due to condyle endochondral ossification, thus correcting the orthodontic anomaly. Their effect though remains controversial. The aim of the present study was to investigate the effect of such functional appliances in the mandible of growing rats by reviewing the existing literature up to March of 2022. Most of them stated that true condylar growth is observed, although there are many limitations due to the nature of such experiments.

**Abstract:**

Skeletal Class II malocclusion is the most common skeletal anomaly in orthodontics. Growth in the body of the deficient mandible is induced by periosteal apposition and endochondral ossification in the condyle. Functional appliances have been used in the correction of Class II malocclusions by inducing mandibular growth. Despite their utilization though, their effect still remains controversial. The aim of the present study is to review the existing literature regarding the effects of mandibular protrusion in mandibular growth of growing rats. A protocol was followed according to the guidelines of the *Cohrane Handbook for Systematic Reviews*. Databases were searched using a specific algorithm. From the ten studies finally analyzed, we conclude that the use of a functional appliance in growing rats induces cell proliferation and bone formation in their condyles, resulting in mandibular growth.

## 1. Introduction

Skeletal Class II malocclusion is the most common skeletal anomaly in orthodontics. Growth in the body of the deficient mandible is induced by periosteal apposition, whereas growth in the ramus is mainly due to endochondral ossification in the condyle [1,2]. According to the literature, there are various environmental factors that could possibly affect mandibular growth, such as diet consistency [3]. Functional appliances have been used in the correction of Class II malocclusions by inducing mandibular growth. Despite their utilization though, their effect still remains controversial as to their long-term effects compared to one-phase orthodontic treatment [4]. Their clinical effectiveness remains a controversial subject due to many reasons. The main reasons are that the amount of true protrusion reported varies among researchers, and the method of measuring it is determined by parameters set by each researcher individually; most of the time, the parameters are not comparable to each other. The last but most important reason of this controversy is that because these appliances are all utilized on human beings, ethics do not permit the conduction of randomized clinical trials [2]. Many researchers conclude that mandibular protrusion with functional appliances induces condyle growth and growth in mandibular length [5,6]. Quite as many though, propose that this is not the case [7]. The different outcomes of functional appliance therapy may also be due to the method, degree and/or duration of mandibular advancement used by each individual researcher [8]. 

Mandibular growth usually follows the general growth curve. This means that the mandible follows the adolescent growth spurt. The peak of this growth spurt, and a little bit before it, is the right time to correct any kind of malocclusion with a skeletal aspect, such as mandibular deficiency. That way, after the peak of growth, any skeletal problems will have been corrected, and only a little more growth at a slower pace will be remaining. At this time, fixed orthodontic appliances can be used to correct any remaining problems [2]. According to some researchers, mandibular propulsive appliances increase cell proliferation, IGF I, IGF II and collagen-binding integrin expression in the condylar cartilage of growing rats [9,10]. Mechanical forces are important for the maintenance of the condylar cartilage. Strenuous loading inhibits cell proliferation [11], while optimal loading raises anabolic response in chondrocytes [12]. Decreased loading is due to the cutting of incisors in experimental animals or to a soft diet, which decreases cell proliferation and extracellular matrix formation [13]. Mechanical forces are transmitted to cells through integrines, which are tansmembrane receptors [11,14]. 

Indian hedgehog (Ihh) is a transcription factor that regulates chondrocyte proliferation and differentiation [15]. It uses mechanical signals induced by forward mandibular positioning to stimulate cellular proliferation in the condyle. Hence the measurement of the expression levels of Ihh is an indicator of mandibular growth during functional appliance treatment [12]. 

Type II collagen has also been found to be increased on the forward positioning of the mandible [16]. Cartilage is the matrix onto which new bone will form. The more cartilage there is, the greater the potential for new bone formation. Therefore, measuring type II collagen during mandibular advancement can indicate the effect it has on bone growth [8]. Type X collagen is the major extracellular component synthesized by hypertrophic chondrocytes in order to be calcified [17]. It provides an easily resorbed fabric for the deposition of the bone matrix, providing support during the degradation of the cartilage [18]. 

Many studies have shown that new bone formation in the condyle and glenoid fossa, occurring due to mandibular advancement, is regulated by fibroplast growth factor 8(FGF8). This process involves endochondral ossification in the condyle and intramembranous ossification in the glenoid fossa. FGF8 regulates mesenchyme and chondrocyte cell proliferation [17,18,19]. 

The aim of the present study is to review the existing literature regarding the effects of mandibular protrusion in mandibular growth in growing rats.

## 2. Materials and Methods

A protocol was followed according to the guidelines of the *Cohrane Handbook for Systematic Reviews*.

### 2.1. Eligibility Criteria

Eligibility criteria were formed according to the PICOS guidelines, as seen in the table below (Table 1). Prospective studies utilizing growing rats were included. Reviews and meta-analyses were excluded.

### 2.2. Information Sources

A literature search was carried out by applying the Medline Database (via PubMed) and Scopus. The following terms were used: ((Mandibular protrusion) OR (Mandibular anterior displacement) OR (Mandibular growth)) AND (Rats).

### 2.3. Search Strategy

Two databases were searched by two independent researchers from inception to March 2022. In addition, reference lists were searched for further studies. The article was registered in PROSPERO and received the ID number CRD42022342750.

### 2.4. Risk of Bias

Risk of bias assessment was analyzed according to the *Cohrane Handbook for Systematic Reviews* [20]. Due to the nature of the research, blinding of performance was not feasible in any of the studies. No data went missing during the experiments, so we considered the studies high quality as far as attrition is concerned. No studies used any kind of blinding as far as result assessment is concerned.

## 3. Results

### 3.1. Article Selection

A total of 1465 articles were found. The author assessed the search results and picked 47 of these according to the title for further assessment. From the 47 articles, only 14 were kept and retrieved after reading their abstracts, keeping in mind the eligibility criteria set prior to the search. From the 14 remaining articles, 4 were excluded at the end. The 10 remaining articles were read and analyzed in order to reach a conclusion (Figure 1).

### 3.2. Study Characteristics

Table 2 shows the characteristics of the included studies. All of the included studies used some kind of functional appliance, with the majority of them using either bite-jumping appliances or inclined planes. The majority utilized histological analysis, and some used radiographic analysis as well, that being either CBCT or X-ray.

From the research papers, we conclude that the use of a functional appliance in growing rats increases cell proliferation in the condylar cartilage. One out of ten articles concluded that functional appliances have no effect on mandibular growth. The effect of mandibular anterior displacement on the mandible seems to be reliant on the hours of application and the amount of protrusion that the appliance makes. 

### 3.3. Risk of Bias within Studies

The risk of bias assessment is outlined in Table 3. 

Six studies used randomization methods when allocating animals into experimental and control groups [8,16,17,21,22,23], although they did not specify the method used. None noted any blinding of the experimental procedure, as this was not feasible with live animals wearing appliances. As far as the result assessment for blinding is concerned, three studies made an effort to eliminate bias, resulting in a low risk in this category [17,23,24]. No data were reported missing in any of the studies, and a reporting bias is of low risk in almost all of the studies since almost all the reported results correspond to the intended outcomes.

### 3.4. Results of Individual Studies

The use of a growth hormone in growing rats in combination with functional appliances results in condylar growth. The same results, but significantly less growth, were reported with the application of functional appliances without administering a growth hormone according to Wang et al. [21]. There is also a correlation between the amount of mandibular advancement and the growth of the condyle. There seems to be a threshold over which condyle growth is generated [8]. Cellular morphology and hypertrophy differentiate during mandibular advancement in the condyle, which undergoes endochondral ossification, and in the glenoid fossa, which undergoes intramembranous ossification [23]. The number of mesenchymal cells present in the mandibular condyle cartilage tissue is closely connected to the number of osteoblasts that form new bone during growth. FGFs are factors that regulate the proliferation of mesenchymal cells and chondrocytes. In particular, FGF8 induces hypertrophy and some morphological changes in chondrocytes. This is the reason why FGF8 is used as an indicator of new bone formation in many research papers [23]. 

Oksayan et al. used a bite-jumping appliance that produced a 3.5mm anterior displacement and a 3mm downward displacement. The appliance resulted in a forward growth of the mandible, but no vertical changes [22]. 

One of the studies using an intermittent appliance for anterior displacement of the mandible for ten hours a day reported no difference in mandibular growth between the control and experimental groups [24]. Another study with intermittent use of such an appliance stated no difference between the control and experimental groups in the first days of the experiment, but noted significant changes in the condyle and glenoid fossa after thirty days of appliance usage [25]. One more study with intermittent use of a mandibular protrusion appliance resulted in changes in the anterior region of the condyle, where cell proliferation was noted [26]. Hajjar et al. utilized an inclined plane on the upper incisors for ten hours a day reporting an increase in IGF I and IGF II and thus a remodeling of condylar cartilage [9]. 

Shen et al. used a bite-jumping appliance and noticed a type X collagen increase and maturation of chondrocytes, all indications of mandibular growth [16]. Rabie et al. concluded the same results, stating that forward mandibular positioning accelerates and enhances chondrocyte differentiation and cartilage formation [27]. 

## 4. Discussion

### 4.1. Summary of Evidence

Mandibular protrusion with functional appliances has been widely used to enhance the growth of the condyle and with it the whole mandible. This growth is influenced by many factors, such as epigenetic and mechanical ones. According to some studies, the gender of the animals used plays an equally important role because estrogen can affect bone density. However, most appliances were used prior to the growth spurt, where estrogen levels are not significantly different between males and females [20]. The results of Class II functional appliance treatment also depend upon the capacity of the bite-jumping appliance [21]. 

Mechanical stress is applied to tissue cells in many different circumstances. Stress is even applied during embryological development, as well as the natural process of growth [21]. It is said that endochondral ossification can occur in the condyle when stress is applied, by an anterior mandibular positioning appliance, for instance. This process begins with an increase in the number of nondifferentiated mesenchymal cells. In order for cartilage to be removed and bone to be produced in its place, new blood vessels have to reach the area and bring blood with them, carrying a number of nondifferentiated mesenchymal cells. These cells potentially turn into osteoprogenitor cells, which in their turn become osteoblasts, thus producing new bone. VEGF is a neovascularization factor that plays a role in this procedure. It is known to be expressed in the condyles of growing rats, as well as in the bone formation in long bones. The amount of expression of VEGF if monitored can relate to the amount of new bone being formed during anterior mandibular displacement. [15]. Endochondral ossification not only includes cellular differentiation, but also the process of creation and resorption of the cartilage matrix [28]. According to this process, mesenchymal cells differentiate into chondrocytes. These chondrocytes are responsible for the production of cartilaginous matrix. As chondrocyte cells become hypertrophic, the matrix is not only remodeled but also calcified. When neovascularization occurs, cartilage is resorbed. Each of these steps of differentiation occurs not only in a precise pattern, but is also regulated by specific molecular regulators [29]. The beginning of endochondral ossification, as mentioned above, is marked by neovascularization. VEGF is known as an essential molecular regulator of the neovascularization step [30]. The expression of VEGF is therefore important in endochondral ossification and is often measured by researchers to show the production of new bone. It is proven that its expression is increased with mechanical strain [31,32]. In culture models with chondrocyte cells, tension has been reported to activate the Cbfa/MMP13 pathway and thus increase the expression of hypertrophy and differentiation markers, such as type X collagen and VEGF. In growing rats, the expression and upregulation of VEGF is translated into new bone formation in condyle and glenoid fossa. This mechanical strain is produced by mandibular advancement devices [33]. Genetic mechanisms involved in fetal skeletogenesis are also responsible for regulating adult skeletal regeneration. Earlier, VEGF was considered to be an endothelial-specific factor based on the exclusive distribution of its tyrosine kinase receptors, VEGFR-1 (Flt-1) and VEGF-2 (Flk-1) in endothelial cells [34,35]. While more recently, accumulating evidence has shown that VEGF expression and VEGF receptors are present in nonendothelial tissues, such as osteoblasts, chondrocytes, trophoblast cells and uterine smooth muscle cells [36]. 

Oksayan et al. studied the radiographic evaluation of bite-jumping appliance effects on rats. The evaluation showed a statistically significant difference in horizontal measurements of the mandible between the experimental and control groups. Vertical changes had no statistical significance though [22]. 

Condylar cartilage thickness also increases due to anterior mandibular displacement. This increase is even greater when a growth hormone is administered to experimental animals [21]. 

### 4.2. Strengths, Limitations of Current Studies and Recommendation for Future Ones

Further investigation is needed to find methods of radiographic evaluation of mandibular growth in rats, as many of the involved researchers utilize histological markers, and few use X-rays or CBCT. No research has examined the relapse of such treatments after the growth spurt of the experimental animals has ended.

The age of the animals used is of maximum importance as we sought studies with animals in their growth spurt. All studies chosen used growing rats except one [27], and even this one reported new bone formation due to continuous orthopedic forces. Most articles use the functional appliances fitted 24 h a day. Something like that is not feasible in a clinical practice; thus, it would be interesting to see how bone responds to intermittent use of such appliances. Only one study by Tsolakis et al. used an appliance that could safely be removed from the experimental animals for several hours a day so they can properly feed, thus referring to intermittent use [37]. Further investigation with such an appliance is needed in order to rule out the duration of treatment, which clinically cannot match that of the experiments in animals.

Another factor that plays an important role in mandibular growth is the duration of appliance usage. No research from the ones found used the appliance for a duration of more than 30 days, with 10 days being the average time of use. It would be interesting to know if covering the whole growth period of rats would make any difference in the outcomes and in their stability.

The most important factor of all is the type of the appliance used in each study. The rat’s anatomy is such that the mandibular resting position is far behind what is expected, and lateral movements of the mandible are very common [38]. Thus, an inclined plane is difficult to achieve as is maintaining a forward positioning of the lower jaw, at least for the time needed to have any marked results [37]. Rats are widely used in functional appliance therapy studies because of the histological similarities between their TMJ and human TMJ, even though there are some morphologic differences [23]. 

Some studies researched the amount of protrusion needed to have bone formation and a significant growth of the mandible. Various degrees of mandibular advancement can produce different mechanical strains and thus different amounts of bone formation. There is the possibility of having to surpass a certain threshold to achieve a response of the bone [8,39]. On the other hand posterior displacement of the mandible in rats affects bone modeling under a minimum strain threshold that controls mandibular growth [40,41].

Some researchers state that there is a close connection between the cartilage matrix and, therefore, cartilage increase and new bone formation, as a result of forward mandibular displacement due to functional appliance use. They therefore concluded that mandibular growth mainly depends on chondrogenesis [16]. 

## 5. Conclusions

From all the above, we could safely conclude that true mandibular bone formation can be achieved through the use of functional appliances for mandibular advancement, although further research is needed to determine the optimal time and duration needed to achieve optimal results.

## Figures and Tables

**Figure 1 animals-12-02059-f001:**
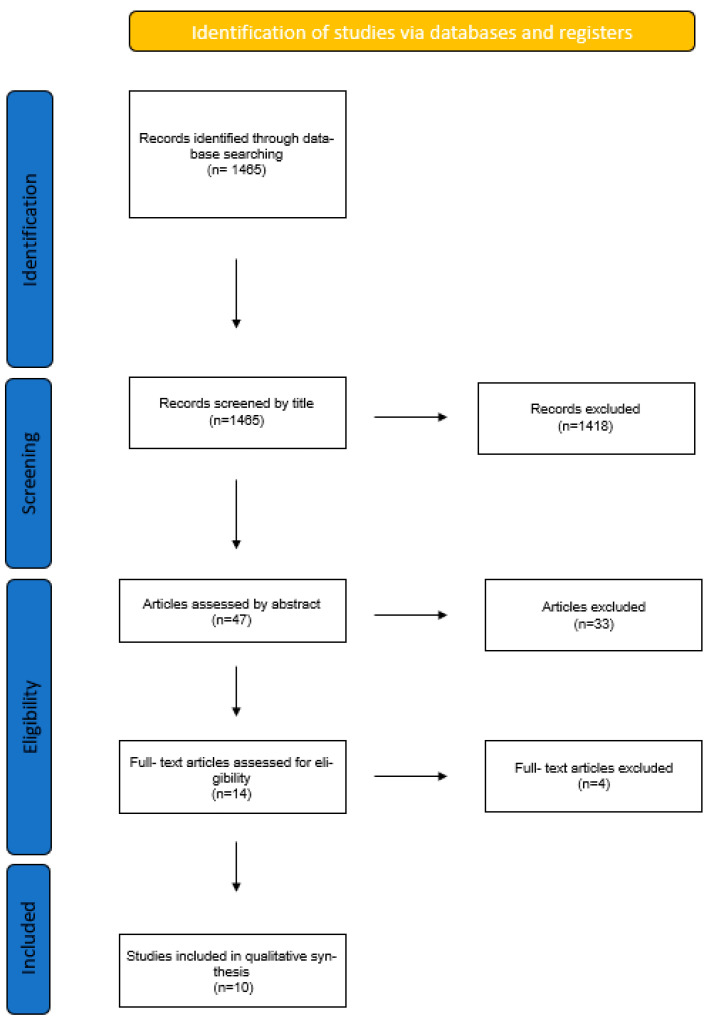
Flowchart of article selection.

**Table 1 animals-12-02059-t001:** Eligibility criteria according to PICOS.

Population	*Growing Rats*
**Intervention**	*Functional appliances*
**Comparison**	*Control group*
**Outcomes**	*Prospective studies*
**Study design**	*No reviews or meta-analyses* *Only anterior movement assessment* *Only in English language*

**Table 2 animals-12-02059-t002:** Studies included in the research.

Article	Sample	Intervention	Method of Assessment	Results
Wang et al. 2018 [21]	40 6-week-old female Wistar rats 4 groups: Control Growth hormone Functional appliance Functional appliance and growth hormone	Functional appliance (inclined plane on upper incisors and occlusal plane on lower incisors) and growth hormone used.	CBCT Cephalometric radiograph Histological analysis	Experimental groups had statistically significant differences in mandibular growth with the control group. No differences among experimental groups.
Rabie et al. 2008 [8]	335 rats Group 1 with 4 mm protrusion Group 2 with 2 mm protrusion Control group no appliance	Bite-jumping appliance: inclined planes bonded on both upper and lower incisors	Immunohistology	4mm protrusion group showed more bone formation than the 2mm group and the control group.
Owtad et al. 2011 [22]	55 24-days-old female Sprague–Dawley rats	Crown former on lower incisors that caused a mandibular forward and downward positioning.	Immunohistology (FFG8)	FFG8 expression greater in condyle and glenoid fossa of experimental group, thus concluding bone formation. Condylar cartilage depicts endochondral ossification, whereas glenoid fossa intramembranous ossification.
Oksayan et al. 2014 [23]	24 8-week-old male Wistar albino rats	Bite-jumping appliance on mandibular incisors (3.5 mm anterior displacement)	Lateral cephalometric X-ray	Increased mandibular and condylar growth but not in the vertical dimension.
Tewson et al. 1988 [24]	114 male 4-week-old Lister Hood and Sprague–Dawley rats	Removable bite plate retainer 10 h a day (2 mm anterior displacement, 3 mm inferior)	Histological analysis Autoradiography	No differences in control and experimental groups.
Tonge et al. 1982 [25]	55 female Lister Hood rats	Cast gold bite plane on upper incisors and stainless-steel mesh with elastics	Histological analysis	Differences observed only 30 days after the beginning of experiment.
Marques et al. 2008 [10]	56 28-day-old male Wistar rats	Inclined plane 6 h a day	Histological analysis Real time PCR	Fibronectin increases 15 days after start of experiment Cell proliferation increases on the anterior region of the condyle
Shen et al. 2006 [17]	100 Sprague–Dawley rats 5 weeks of age	Bite-jumping appliance on upper incisors (3.5 mm forward positioning)	Histological analysis	Condylar forward positioning results in enhanced maturation of chondrocytes and increased type X collagen synthesis.
Rabie et al. 2003 [26]	160 5-week-old female Sprague–Dawley rats	Bite-jumping appliances on upper incisors	Histological analysis	Forward mandibular positioning accelerates and enhances chondrocyte differentiation and cartilage formation.
Hajjar et al. 2003 [9]	70 21-day-old Wistar rats	Inclined plane on upper incisors 10 h a day	Histological analysis	Increase of IGF I and IGF II showing their important role in cell differentiation and remodeling of the mandible.

**Table 3 animals-12-02059-t003:** Risk of bias. Each study was assessed according to signaling questions: Was the sample randomly selected? Was the experiment procedure blinded? Was the result assessment blinded? Did any data go missing during or at the end of the experiment from any group? Is there a reporting bias?

Studies	*Selection*	*Performance*	*Detection*	*Attrition*	*Reporting*
Wang et al. 2018 [21]	Low	High	High	Low	Low
Rabie et al. 2008 [8]	Low	High	High	Low	Low
Owtad et al. 2011 [23]	Low	High	Low	High	Low
Oksayan et al. 2014 [22]	Low	High	Unclear	Low	Unclear
Tewson et al. 1988 [24]	High	High	Low	Unclear	Low
Tonge et al. 1982 [25]	High	High	High	Low	Unclear
Marques et al. 2008 [10]	Unclear	High	High	Low	Low
Shen et al. 2006 [17]	Low	High	Low	Low	Low
Rabie et al. 2003 [26]	Low	High	High	Low	Low
Hajjar et al. 2003 [9]	High	High	High	Low	Unclear

## Data Availability

The data presented in this study are available in the included studies of this systematic review.

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
