# Peer review of "Anterior Mandibular Displacement in Growing Rats—A Systematic Review"

_animals, 2022, doi:10.3390/ani12162059_

Round 1

Reviewer 1 Report

"The last but most important reason of this controversy is that because these appliances are all utilized on human beings, ethics do not permit the conduction of randomized clinical trials". Why? I can understand this logic if the appliances involved are new or experimental. However, concerning the appliances existed in the market or approved by the authorities, why RCT cannot be done to test which one of them is more efficient for a specific group of population?

For risk of bias, majority of the 10 studies had High risk for Performance and Detection. The authors should further elaborate on why this happened, and suggest ways for future studies to improve.

In the Methods it was reported that both PubMed and Scopus were searched. However in the Results it was reported that only PubMed was searched. Please clarify, and report the number of articles yielded from PubMed and Scopus, respectively.

Table 3 is actually a figure, and it is not cited in the main text. Please correct it.

Author Response

  1. The appliances mentioned here all are based on the same principle, the anterior displacement of the mandible. So in this case we are testing the whole concept of anterior mandibular displacement. They are widely used in orthodontics but controversy remains as to whether they impose a true skeletal effect or mainly a change in function. Rcts in this case would be conducted on human beings that have a Class II skeletal malocclusion. In these cases we will need to use appliances that displace the mandible anteriorly. Studies in this case would need an experimental and a control group. The latter will receive a placebo or no therapy. Leaving a group of people with a diagnosed skeletal problem, without treatment cannot be done since the age span in which such an anomaly can be corrected with functional appliances is between eight and ten years old, and after that they cannot be corrected.

  2. Performance and detection bias are seen on 10 studies because of the nature of the experiments conducted. There is no way to be blinded as to which animal belongs into the experimental or control group because the appliances used are visible. During measurements however, a certain degree of blinding is feasible.

  3.  This was a typo. The search results are from both pubmed and scopus and it is corrected in the text. Thank you for your comment.

  4. Thank you about your comment. We have proceeded to change it in the main text.

Reviewer 2 Report

I have two concerns regarding the paper 

1. In the introduction the statement "Growth in the body of the deficient mandible is induced by periosteal apposition, whereas growth in the ramus is mainly due to endochondral ossification in the condyle [1,2]." I do not believe this is true for human mandibular growth. I am not sure about rats. 

2. The conclusion "From the ten researches finally analyzed we conclude that the use of functional appliance in growing rats induces cell proliferation and bone formation in their condyles, resulting in mandibular growth." I do not think you can say anything more than "cell proliferation was increased in the condylar cartilage" which might lead to increased mandibular growth but it also might not.

Author Response

  1. Thank you about your comment and your help. Periosteal bone apposition is a main point where the whole philosophy of the frankel appliances is based. Enlow (AJO 1964) talked about remodeling and repositioning of the mandible. The mandible grows in several principle directions simultaneously. As growth of all local areas proceeds in their particular directions the various parts of the mandible necessarily become relocated into various relative positions. As an area changes its relative location in the mandible, the bone in this area is partially or totally removed and some new deposits are added onto old surfaces to bring about the local adjustments needed. A major site of growth in the mandible is the condyle that undergoes endochondral ossification, but this happens in the whole body of the mandible through intramembranous ossification.

  2. We thank the reviewer for his comment. The above has been changed in the main text accordingly.